# Effect of Glycerol as Processing Oil in Natural Rubber/Carbon Black Composites: Processing, Mechanical, and Thermal Aging Properties

**DOI:** 10.3390/polym15173599

**Published:** 2023-08-30

**Authors:** Weerawut Naebpetch, Sutiwat Thumrat, Yeampon Nakaramontri, Suppachai Sattayanurak

**Affiliations:** 1Department of Rubber and Polymer Engineering, Faculty of Engineering, Thaksin University, Phatthalung Campus, Phatthalung 93210, Thailand; weerawut.n@tsu.ac.th (W.N.); sutiwat.thumrat43@gmail.com (S.T.); 2Advanced Material Research Center, National Research and Innovation Agency (BRIN), KST B.J. Habibie, Serpong 15314, Indonesia; indr017@brin.go.id; 3Sustainable Polymer & Innovative Composite Materials Research Group, Department of Chemistry, Faculty of Science, King Mongkut’s University of Technology Thonburi, Bangkok 10140, Thailand

**Keywords:** glycerol, natural rubber, processing oil, carbon black, mechanical properties

## Abstract

The present work aims to study the effect of glycerol as a replacement for mineral oils in natural rubber (NR) composites to obtain suitable properties via cure characteristics, mechanical properties, and thermal stability. Glycerol was used at a 5 phr rate in the compound with carbon black as a reinforcing filler and was compared to mineral processing oils such as aromatic oil, treated distillate aromatic extracted oil, and paraffinic oil. Compared to the other oils, glycerol exhibits better maximum torque and torque differences. Also, a shorter scorch time, cure time, and a higher cure rate index of the compounds were observed. However, although the received mechanical properties, including tensile strength, elongation at break, and compression set of the vulcanized rubber using glycerol showed slightly lower values than the others, the 100% and 300% moduli, as well as the hardness of the composites filled with glycerol, exhibit better values relative to the other commercial oils. These findings demonstrate that glycerol overall presents a good balance of properties, making it beneficial to use glycerol as a substitute for mineral oil in tire, shoe sole, and rubber stopper applications.

## 1. Introduction

Natural rubber (NR) is a non-polar hydrocarbon which contains only carbon and hydrogen atoms in its chemical molecular structure; therefore, NR is highly soluble in non-polar solvents such as benzene, hexane, and others. NR generally has an amorphous structure, while under some conditions, the rubber molecules can become relatively organized at low temperatures or when stretched, so crystals can be formed. Crystallization makes the rubber harder, while crystallization due to elongation (strain-induced crystallization) provides the rubber with excellent mechanical properties like tensile strength, tear resistance, and high abrasion resistance. The disadvantage of NR is that it deteriorates quickly under sunlight, oxygen, ozone, and heat, while NR molecules have an abundant amount of double bonds in the main carbon chain, and so, NR is sensitive to oxygen and ozone reactions with sunlight and heat as the catalytic media. Therefore, it is required to add anti-degradants in rubber formulations in order to prolong the service life of rubber products. However, aside from anti-degradants, other additives also need to be added to rubber formulations, such as fillers, processing oils, accelerators, vulcanizing agents, activators etc., in order to achieve the required properties.

Carbon black (CB) is the most widely used and important reinforcing filler material for rubber compounds in order to improve the tensile and tear strength, tensile modulus, tire skid and abrasion resistance, and other properties [1]. CB is therefore the main reinforcing agent in the rubber field [2,3,4]. The properties of the reinforced material are affected by the filler’s particle size, specific surface area, structure, and surface chemistry [1,5]. CB is also used as a coloring agent in the rubber field, especially in tire applications for reason of its chemical and thermal stability [6]. Mixing CB or other filler types in rubber compounds faces several problems, such as that the fillers do not disperse well, are difficult to mix at high loadings, and fillers tend to re-agglomerate, thereby causing a decrease in the physical and mechanical properties of NR [7]. Therefore, processing oil is necessary to be added in rubber compound formulations in order to solve these problems. The function of adding processing oils is to reduce the required energy and viscosity of the rubber compound during mixing and thus improve the processability. The compatibility and suitability of the types of processing oils are important factors and need to be carefully considered in rubber formulations.

Processing oil plays some important roles in rubber formulations. It can be used in rubber compounds in order to improve processability, tackiness, filler dispersion, the flow of the uncured compounds, and the low-temperature properties of the cured compounds [8]. Processing oil can act as lubricant between the polymer chains to improve flow. The compatibility of processing oil with rubber is an important parameter to achieve a proper balance in properties. Non-polar oils are generally used in non-polar rubbers, whereas polar oil is generally used in polar rubber. Processing oil can be generally classified as mineral oil (petroleum-based), synthetic oil, vegetable oil, and other natural oils. Mineral oils are further classified via their relative contents of aromatic, paraffinic, and naphthenic hydrocarbons. All mineral oils are non-polar and relatively cheap, so they are widely used in non-polar rubbers such as NR, polyisoprene (IR), polybutadiene (BR), styrene butadiene rubber (SBR), butyl rubber (IIR), and ethylene propylene diene rubber (EPDM). Processing oils are commonly used in small amounts, mostly 5 to 20 parts per hundred rubber (phr). The major drawback of petroleum-based oils is that they are derived from non-renewable resources and that their aromatic content requires labelling them as carcinogenic. Replacing mineral oils with vegetable oils or other products may be advantageous as a solution to these problems.

Glycerol (or glycerine, glycerin) is a simple polyol compound that can be produced from fats and oils and is a by-product of the production of biodiesel. Therefore, it is produced on a large scale annually worldwide, at very low cost. Moreover, it is a nontoxic, biocompatible, and polyhydric alcohol [9]. The most important factor for being advantageously employed in the present research is its high boiling point of 290 °C [10]. For this reason, it can be used in a reactive system, showing very low mass loss due to evaporation at temperatures below 290 °C. It bears the additional advantage of having a high reaction activity attributed to little steric hindrances. Generally, glycerol is used as a plasticizer in food technology where it can reduce intermolecular forces and the glass transition temperature (Tg), and can increase the chain mobility of proteins [11,12]. Moreover, glycerol finds application in various industrial sectors, including the pharmaceutical industry, cosmetics, and personal care items. It serves as a humectant in food, contributes to the production of resins, detergents, plastics, and tobacco, and also functions as a plasticizer [13]. The effect of glycerol on a NR/wheat gluten (WG) blend was investigated. The results showed that the dispersion of WG was improved by adding glycerol to the rubber matrix. It also affected the cure characteristics and improved the mechanical properties of NR filled with WG [12]. A maximum tensile strength of 23 MPa and rebound resilience of 71% were achieved for CB and the glycerol loadings at 50 phr and 7 phr, respectively, for the glycerol-modified NR/starch-filled CB composites [14]. Calcium lignosulfonate was introduced into styrene–butadiene rubber (SBR) and acrylonitrile–butadiene rubber (NBR) at a consistent ratio of 30 phr. Glycerol, utilized as a rubber-formulation plasticizer, varied in concentration from 5 to 20 phr. The investigation demonstrated that incorporating glycerol as a plasticizer led to a decrease in the viscosity of the rubber compounds, facilitating improved dispersion and even the distribution of the filler within the rubber matrices. This, in turn, enhanced the adhesion and compatibility between the filler and the rubber matrices, resulting in a notable enhancement of tensile properties [13]. An array of compounds derived from glycerol were investigated for their potential as plasticizers in flexible poly(vinyl chloride) (PVC) blends. The influence of the chemical structure of the plasticizers on their performance, migration tendencies, and blend morphology was assessed and juxtaposed with blends employing the commercial plasticizer dioctyl terephthalate (DOTP). Blends containing 40 phr (parts per hundred rubber) of glycerol-based plasticizer exhibited a significant reduction (ranging from 54 to 86 °C) in glass transition temperature (Tg) when compared to pure PVC (with a Tg of approximately 80 °C). Tensile testing of the samples formulated with the glycerol-derived alternatives revealed higher flexibility (elongation at break values reaching up to 97%), in contrast to DOTP (with an elongation at break value of 75%) under the equivalent plasticizer contents [15]. The hydrophobic surface characteristics of silicone rubber was enhanced via electron beam irradiation while utilizing a glycerol layer. Despite glycerol’s intrinsic hydrophilic nature, the treatment resulted in a surprising 19.9% increase in the contact angle of the silicone rubber. This enhancement in hydrophobicity can be attributed to the formation of a network structure on the silicone rubber surface, despite the notable hydroxyl content [16]. The decrease in the elastic modulus as the dielectric permittivity rises, combined with the escalating glycerol loading, was observed in glycerol-filled silicone rubber composites, indicates the suitability of this material as a prospective option for soft dielectric sensors and actuator applications [17,18].

In the present research, the effect of glycerol as a processing oil on the properties of CB-filled NR compounds is compared with different types of processing oils such as aromatic oil, paraffinic oil, and treated distillate aromatic extracted oil. For the formulations involving CB, this filler and the processing oils are added in amounts of 30 and 5 phr, respectively. The processing properties, mechanical properties, and thermal aging resistance are investigated in order to elucidate the effect of glycerol as a replacement of the other processing oils.

## 2. Experimental and Characterization

### 2.1. Materials

Natural Rubber (Ribbed Smoke Sheet, RSS3) was purchased from Thongthai (1956) Co., Ltd., Bangkok, Thailand. Carbon black N330 grade was purchased from Thai Tokai Carbon Product Co., Ltd., Bangkok, Thailand. Glycerol was purchased from Wako Pure Chemical Industries Ltd., Osaka, Japan; the physical appearance of glycerol is illustrated in Figure 1. The other ingredients: Zinc oxide (White seal grade, Utids Enterprise, Co., Ltd., Bangkok, Thailand), stearic acid (Imperial, Co., Ltd., Bangkok, Thailand), Aromatic oil Tudalen 84 grade (H&R ChemPharm (Thailand) Co., Ltd., Bangkok, Thailand), treated distillate aromatic extracted (TDAE) oil Vivatec 500 grade (H&R ChemPharm (Thailand) Co., Ltd., Thailand), paraffinic oil Tudalen 13 grade (H&R ChemPharm (Thailand) Co., Ltd., Thailand), N-Cyclohexyl-2-Benzothiazol Sulfenamide (CBS) (Kawaguchi, Co., Ltd., Tokyo, Japan), and sulfur (Siam Chemical, Co., Ltd., Bangkok, Thailand) were of commercial-grade quality for the rubber industry.

### 2.2. Compound Preparations

Natural rubber filled with CB compounds with different types of processing oil, i.e., aromatic oil, treated distillate aromatic extracted oil, paraffinic oil, and glycerol, are shown in Table 1. The abbreviation of each compound is shown in Table 2. Each compound was prepared via a one-step mixing procedure on a two-roll mill as shown in Table 3. The initial temperature setting of the two-roll mill was at room temperature, and the speed of the front roll and the rear roll were 24 and 33.6 rpm, respectively, with a friction ratio of 1:1.4. First, RSS3 was masticated for 2 min, then ZnO and stearic acid were added in the second step with continuous mixing for 7 min, then CB (N330) and the processing oil were added in the third step with continuous mixing for 18 min, and then CBS and sulfur were added in the last mixing step with continuous mixing for another 6 min.

### 2.3. Sample Characterizations

*Mooney viscosity*—Mooney viscosity, ML(1+4), 100 °C was tested using a Mooney viscometer (EKT2003M, Ektron Tek Co., Ltd., Changhua, Taiwan) according to ASTM D1646 [19].

*Cure/vulcanization characteristics*—The cure/vulcanization characteristics of each compound (i.e., one sample per formulation) were measured by using a Moving Die Rheometer, MDR (MDR-U6S, U-Can Dynatex Inc., Taichung, Taiwan) at a set temperature of 150 °C. The cure rate index (CRI) was calculated as [100/(optimum cure time − scorch time)], with the optimum cure time defined as t_c90_ and the scorch time defined as t_s2_.

*Tensile properties*—The compounds were vulcanized to their optimum cure time (t_c90_) by using a Hong Yao Thai laboratory compression press at 150 °C and 15 MPa pressure into 2 mm thick sheets. For the tensile properties, the dumb-bell test specimen (i.e., five samples per formulation) were die cut from the press-cured sheets, and tests were carried out at room temperature with a tensile tester Model UT-2060 (U-Can Dynatex Inc., Taiwan) at a crosshead speed of 500 mm/min according to ASTM D412 [20].

*Hardness*—Test specimen in the shape of a square, with a size of 6 × 6 cm^2^ and a thickness of 9 mm, were measured according to ASTM D2240 [21] Shore A with a hardness tester Model GX-02 (Teclock Co., Ltd., Nagano, Japan). A Type A probe was used by measuring at the four corners of the test piece and the center of the test piece and saving the measured value. The average value of the five measurements was reported.

*Compression set*—At least three cylindrical test samples, according to ASTM D395 [22], Type A, were used with a diameter of 29.0 ± 0.5 mm and a height of 12.5 ± 0.5 mm. The test samples were incubated in a hot air circulating oven at 70 °C for 22 h. The test samples were allowed to cool at room temperature for 30 ± 3 min before measuring the final thickness of the test samples. The compression set was calculated as [(Initial thickness − final thickness)/(Initial thickness × 25%)] × 100. The average of the three samples was reported.

*Thermal aging resistance*—Dumbbell test samples according to ASTM D573 [23] were aged at 70 °C for 168 h in an aerated UN110 Memmert hot air oven. After cooling, the test pieces were tested for tensile properties at room temperature (i.e., five samples per formulation). The change in properties was expressed as relative to those before aging.

## 3. Results and Discussion

### 3.1. Mooney Viscosity and Cure Properties

The Mooney viscosity and minimum torque (M_L_) of the NR compounds with different types of processing oil are given in Figure 2. Both the Mooney viscosity and M_L_ are indicative of the rubber compounds’ viscosity or flowability. A lower viscosity commonly means an improvement in processability. The viscosity of a rubber compound depends on the compatibility between the rubber and the filler, and the plasticizing effect of the processing oil. In general, fillers that are poorly compatible with rubber usually show a higher viscosity and are difficult to mix. The processing oil is a factor that has an effect on the compatibility between the rubber and the filler. A processing oil which has a similar solubility parameter like that of the filler and the rubber results in a good compatibility with both. Processing oil does improve the dispersion of fillers in the rubber matrix with an effect of lower Mooney viscosity and M_L_ of the rubber compounds. The use of glycerol as a processing oil shows the highest Mooney viscosity and M_L_ compared to the compounds with TDAE, paraffinic, and aromatic oils, respectively. The highest Mooney viscosity and M_L_ for glycerol is due to the high polarity or solubility parameter of glycerol compared to the oil types. Based on the literature, the solubility parameters of the rubber and processing oils are shown in Table 4. The solubility of paraffinic oil, TDAE oil, and aromatic oil are closer to the solubility parameter of NR than glycerol, resulting in a lower Mooney viscosity and M_L_ than using glycerol. This is consistent with the research of Li, X., et al. (2016) [24], which described the effect of the solubility parameter of paraffinic oil and aromatic oil on the Mooney viscosity of IIR compounds. They found that the Mooney viscosity of butyl rubber/paraffinic oil was lower than IIR/aromatic oil. The reason was that the solubility parameter of paraffinic oil at 16.6 MPa^1/2^ [24] was similar to that of IIR with a solubility parameter of 17.2 MPa^1/2^ [24,25,26], compared to the solubility parameter of aromatic oil at 18.4 MPa^1/2^ [24]. In addition, the determination of the improved properties in the composites must consider not only the differences in polarity and solubility between NR and glycerol, but also the compatibility of glycerol with the filler surfaces. Therefore, the utilization of glycerol can also be applied, along with the incorporation of polar fillers, to mitigate the re-agglomeration of fillers after the mixing process, in accordance with the thermodynamic theory’s effects.

Figure 3 shows the cure characteristics of the NR compounds with different types of processing oil. The results demonstrate that all three mineral oils as processing aids have little mutual effect on the cure characteristics of the rubber compounds. On the other hand, the use of glycerol as a processing oil provides a faster vulcanization reaction of the rubber compound: the cure curve of the rubber compound with glycerol shows a clearly reduced scorch time and a faster cure.

The maximum torque (MH) and torque difference (M_H_−M_L_) of the NR compounds with different types of processing oil are shown in Figure 4. The increases in M_H_ and M_H_ −M_L_ are related to a higher degree of crosslinking, though the higher Mooney viscosity also may have had a small contribution. If the M_H_ and M_H_−M_L_ value is high, it indicates that the rubber compound is strong and has a high crosslink density between the rubber molecules, which results in better mechanical and aging properties. As can be seen in Figure 4, the compound using glycerol shows the highest M_H_ and M_H_−M_L_ together with TDAE-oil, compared to the aromatic and paraffinic processing oils. Since the same type of CB is employed, the rubber–filler interaction depends on the secondary structure of aggregates or agglomerates of the CB, the filler dispersion, and the surface activity of the blacks [1,5,28]. As the glycerol compound shows an increase in both the viscosity (ML1+4, 100 °C and M_L_), as shown in Figure 2, and subsequently a higher crosslink density (the increase in M_H_ and M_H_−M_L_), it may be taken as evidence that the interaction between the rubber and the filler is positively influenced.

Figure 5a shows the scorch times (t_s2_) and cure times (t_c90_) of the CB-filled NR compounds with different types of processing oil. The compound with aromatic oil shows a higher t_s2_ and t_c90_ than the TDAE oil and paraffinic oil, while in the case of glycerol, by far the shortest values are found. These lowest t_s2_ and t_c90_ mean better productivity of the rubber products because the output of the rubber products per hour may be increased. The lowest values of t_s2_ and t_c90_ are most likely due to an effect of the polar hydroxyl groups (-OH) of glycerol, which exercise an influence on the vulcanization reactions via hydrogen bonding and blocking the surface hydroxyl groups of the CB filler as widely employed to reduce the accelerator absorption [1,28,29,30,31]. A possible chemical reaction of CB and glycerol is illustrated in Figure 6, modified from the previous work of Al-Juothry, S.A [32]. The hydroxyl groups from glycerol interact with the hydroxyl groups on the CB surface and blocks the absorption of accelerators.

The vulcanization rate index or cure rate index (CRI) is a value that indicates the rate of the vulcanization reaction of the rubber. A high CRI means that the rubber can vulcanize quickly. The effects of the various types of processing oil on the cure rate index (CRI) are shown in Figure 5b. The compound using glycerol shows the highest CRI, as anticipated on the basis of the shortest t_s2_ and t_c90_, again, since the polar OH groups of glycerol prevent accelerator adsorption on the CB-surface and so accelerates the curing reaction. The increase in CRI again shows a positive effect on the production process of rubber products due to the fast vulcanization rate, resulting in an increase in the number of rubber products per hour.

### 3.2. Mechanical Properties

Figure 7 shows the modulus at 100% and 300% elongation of vulcanized rubber with different types of processing oil. The use of glycerol as a processing oil gave the moduli at the 100% and 300% elongations just a bit higher than the other processing oils. This may be the result of glycerol that, besides acting as a processing oil and influencing the CB dispersion, it also helps to condition the surface of CB via hydrogen bonding, and so blocks the surface hydroxyl groups of the CB filler to adsorb the accelerators. The results point in the same direction as t_s2_ and t_c90_, as shown in Figure 5a. It agrees with what was reported previously for NR filled with wheat gluten (WG) and glycerol, that the reinforcement of vulcanized NR/WG increased with the addition of glycerol due to the plasticization effect of WG via the addition of glycerol [12].

The reinforcement index or the ratio between the 300% modulus and the 100% modulus values is shown in Figure 8. The reinforcement index is a measure of the strength and crosslink density in rubber compounds, where it is employed largely in the automotive sector. The types of processing oil do not affect the reinforcement index substantially; they are all more or less the same.

Table 5 shows the tensile strengths, elongations at break, shore A hardnesses, and compression sets of vulcanized rubber using different processing oils. Within experimental error, the tensile strengths, elongations, and hardnesses of the three compounds with mineral-based oils are practically the same. Only the compound with glycerol gives just a little bit lower values for the tensile properties, and similarly, a little bit higher for hardness. This may again be the result of a slightly higher filler–filler interaction, as observed in the highest Mooney viscosity in Figure 2 as well. While the compatibility of aromatic oil, treated distillate aromatic extracted oil, and paraffinic oil was better than glycerol, it resulted in the better tensile strength and elongation at break.

Shore A hardness of vulcanized rubber is also a property that indicates the level of the crosslink density in a rubber compound. A rubber vulcanizate with a higher hardness usually results from an increased amount of crosslinking in the rubber matrices. Table 5 shows the hardness values of vulcanized rubbers using the different types of processing oil. Using glycerol as a processing oil in the compounds shows a little higher hardness than the other processing oils. This corresponds to the perceived torque differences showing in Figure 4, since the torque differentiates from the rubber compounds and is realized as the estimated crosslink density of the compound after vulcanization formation.

Table 5 also shows the compression set values of vulcanized rubber using the different types of processing oil. The worst retainable elastic properties of a rubber vulcanizate are indicated by a higher compression set, while the lower the compression set percentage, the more retainable the elastic properties. As seen, the types of processing oil have little or no effect on the compression sets of the rubber vulcanizates. The main factors affecting the compression set value of rubber vulcanizates are the type of rubber and crosslink density. The glycerol compound showed a higher M_H_ and M_H_−M_L_ than the other types of processing oil, which reflects here on the lowest compression set.

### 3.3. Thermal Aging Properties

The thermal aging of rubber vulcanizates depends on the testing conditions and the crosslink structure. Important environmental factors include the concentration of oxygen and/or ozone, aging time, and aging temperature [33]. Figure 9 shows the percentage change in tensile strength and the elongation at break of the vulcanized rubber compounds using different processing oils after thermal aging. It shows that the lowest percentage of change in tensile strength and the elongation at break were observed for the aromatic oil compound and that the rubber vulcanizates for all types of processing oil after thermal aging were lower than before. In practice, the type of rubber and crosslink density are the most important factors which affects the aging properties. In this case, the NR, which contains carbon–carbon double bonds in the main chain, was used as the main rubber matrix. The thermos-oxidative degradable can easily occur, causing lowering thermal properties after providing aging.

Figure 10 shows the modulus at 100% and 300% elongation of vulcanized rubber with different types of processing oil before and after thermal aging. Contrary to the tensile strengths and elongation at break, it can be seen that the 100% and 300% modulus values of all compounds with different types of processing oil after thermal aging were better than before. This is commonly due to additional crosslinking taking place during the thermal aging process. Moreover, comparing the before and after thermal aging of the compounds using different types of processing oil, the glycerol compound showed the highest 100% and 300% modulus. This result corresponds to the results of the 100% and 300% modulus as shown in Figure 7. The elevated temperature increases the stiffness of the polymers due to the raised levels of crosslinking [34]. This result reinforces the observation of the somewhat reduced tensile strengths and elongation at break, which commonly decreases with the increasing modulus.

## 4. Conclusions

This research is to quantify the effects of glycerol as a processing oil compared with the other types of processing oil such as aromatic oil, treated distillate aromatic extracted oil, and paraffinic oil, for carbon black-reinforced natural rubber compounds. The results can be summarized as follows:-For the processing properties: the use of glycerol shows a higher Mooney viscosity and minimum torque (M_L_), higher maximum torque (M_H_), torque difference (M_H_−M_L_), and cure rate index (CRI) of the rubber compounds, while scorch time (t_s2_) and cure time (t_c90_) are lower than for the other types of processing oil.-For the mechanical properties: the tensile strength, elongation at break, and compression set of the vulcanized rubber using glycerol show just slightly lower values than for the other types of processing oil. Whereas the 100% modulus, 300% modulus, and hardness of the glycerol compound shows slightly higher values than for the other types of processing oil.-For the thermal aging properties: the tensile strength and elongation at break of all samples after thermal aging were lower than before, while the 100% and 300% modulus of the samples after thermal aging were all better than before. Glycerol took a middle position.-The overall results show a slightly reduced processability for glycerol as a processing oil, but a clear improvement of the overall mechanical and good aging properties in comparison with the mineral oil-based processing oils renders this abundantly available chemical a good substitute for many rubbers uses like tires, shoe soles, and rubber stopper applications.

## Figures and Tables

**Figure 1 polymers-15-03599-f001:**
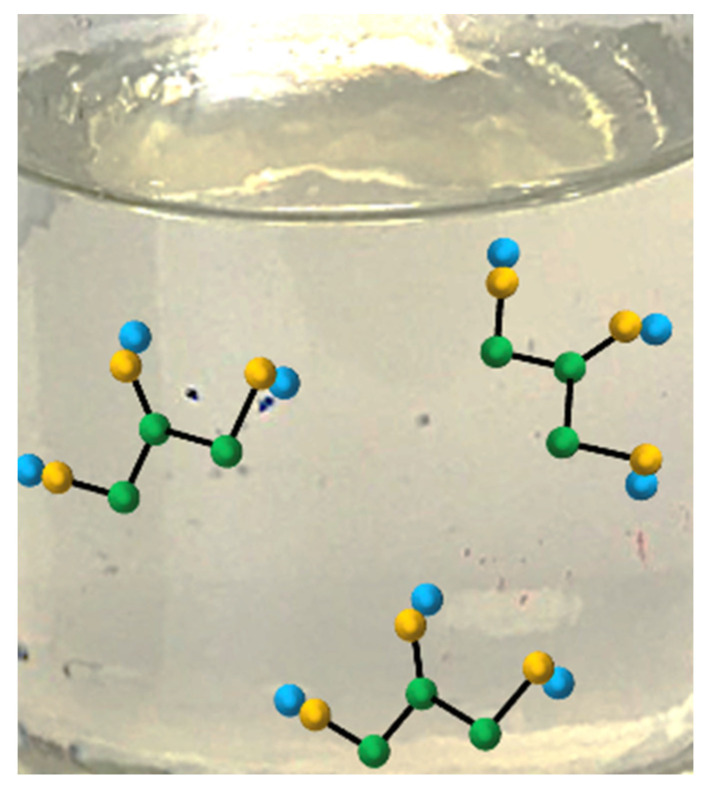
Physical appearance of the glycerol with proposed molecular structure, where green, yellow, and blue cycles refer to carbon, oxygen, and hydrogen atoms, respectively.

**Figure 2 polymers-15-03599-f002:**
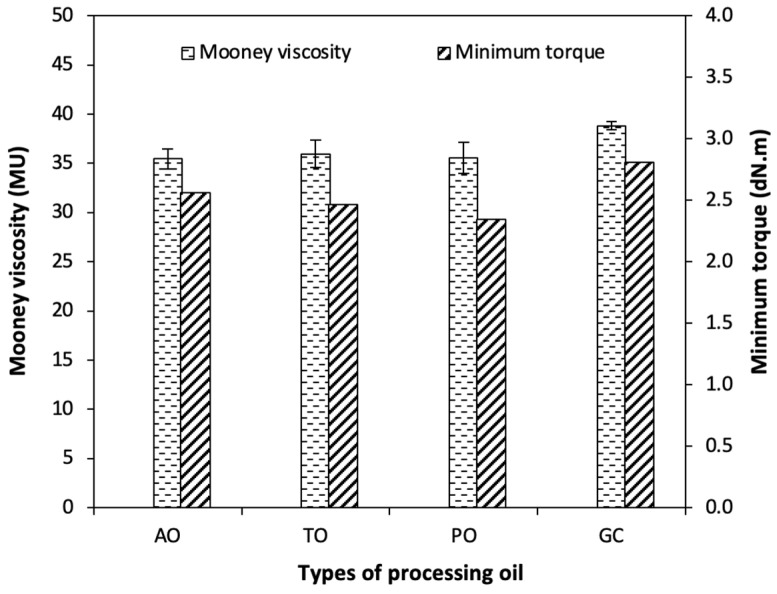
Mooney viscosity and minimum torque of the CB-filled NR compounds with different types of processing oil.

**Figure 3 polymers-15-03599-f003:**
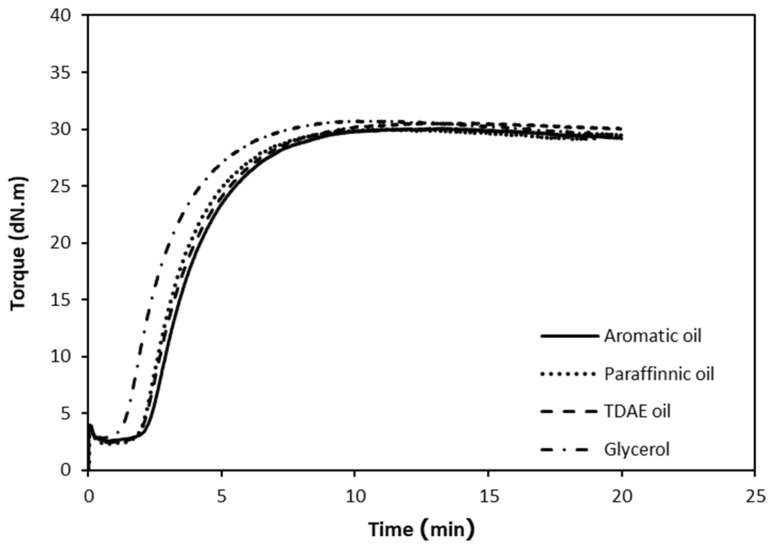
Cure characteristics of the CB-filled NR compounds with different types of processing oil.

**Figure 4 polymers-15-03599-f004:**
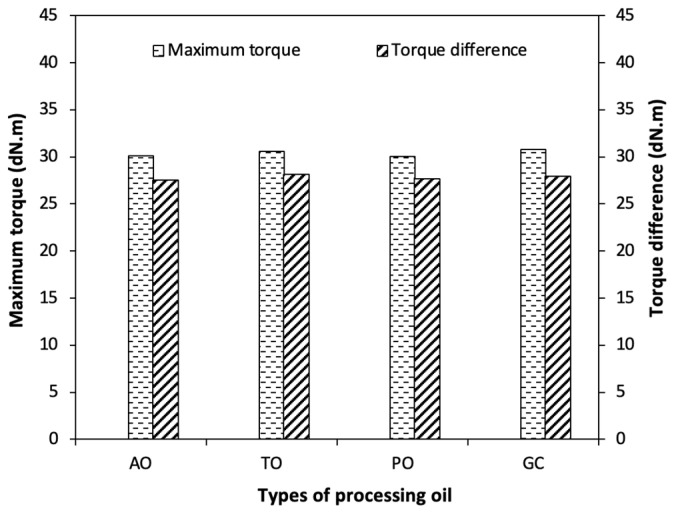
Maximum torque and torque difference of the CB-filled NR compounds with different types of processing oil.

**Figure 5 polymers-15-03599-f005:**
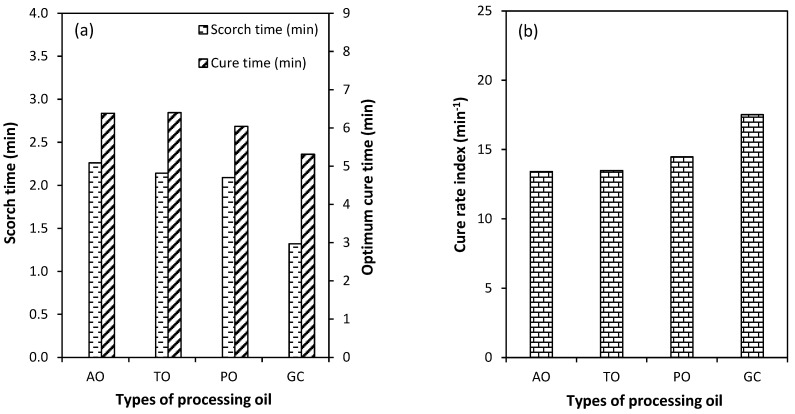
Scorch time and optimum cure time (**a**) and cure rate index (**b**) of the CB-filled NR compounds with different types of processing oil.

**Figure 6 polymers-15-03599-f006:**
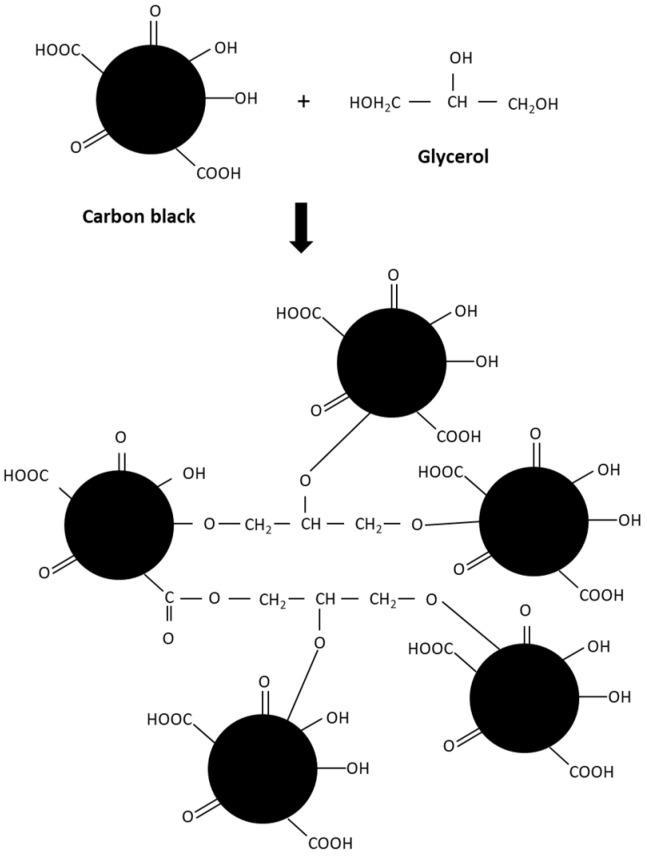
Possible chemical reaction of carbon black and glycerol.

**Figure 7 polymers-15-03599-f007:**
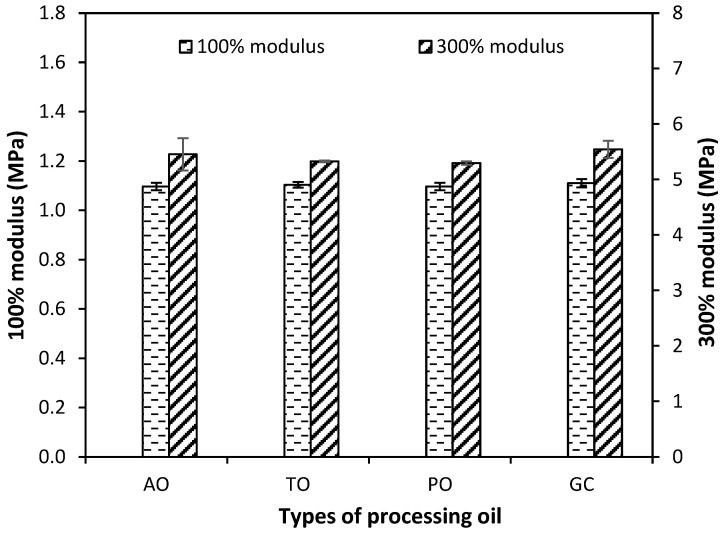
100% and 300% modulus of the CB-filled NR compounds with different types of processing oil.

**Figure 8 polymers-15-03599-f008:**
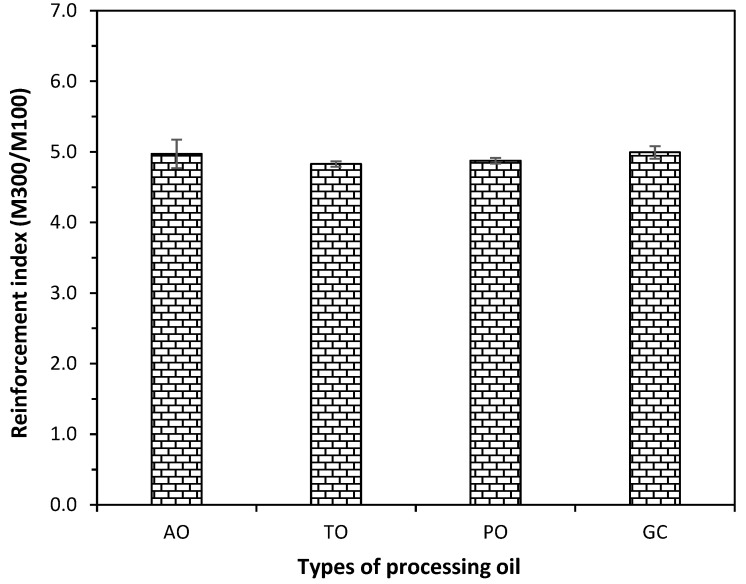
Reinforcement index of the CB-filled NR compounds with different types of processing oil.

**Figure 9 polymers-15-03599-f009:**
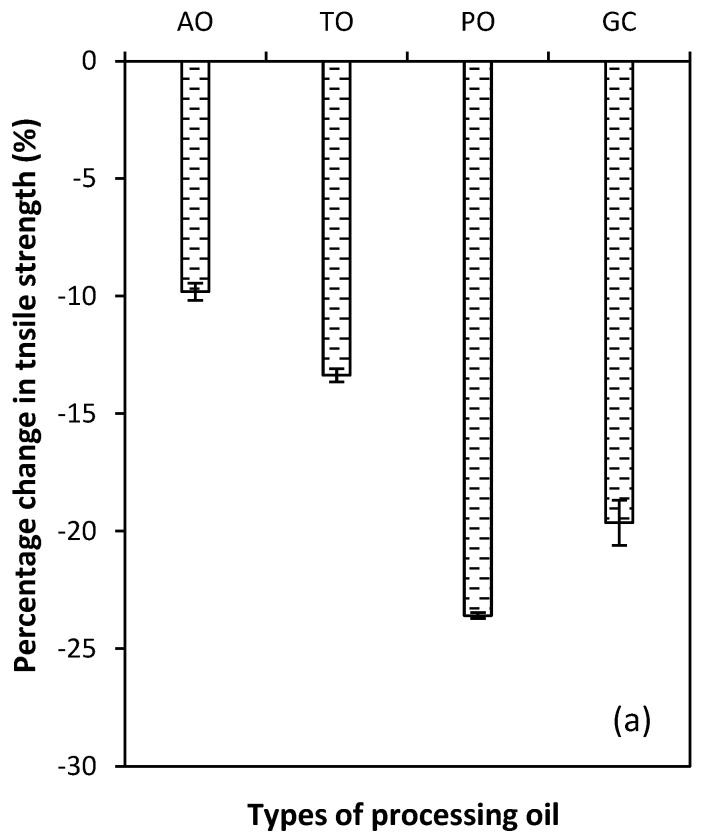
Percentage change in tensile strength (**a**) and elongation at break (**b**) of the CB-filled NR compounds with different types of processing oil after thermal aging.

**Figure 10 polymers-15-03599-f010:**
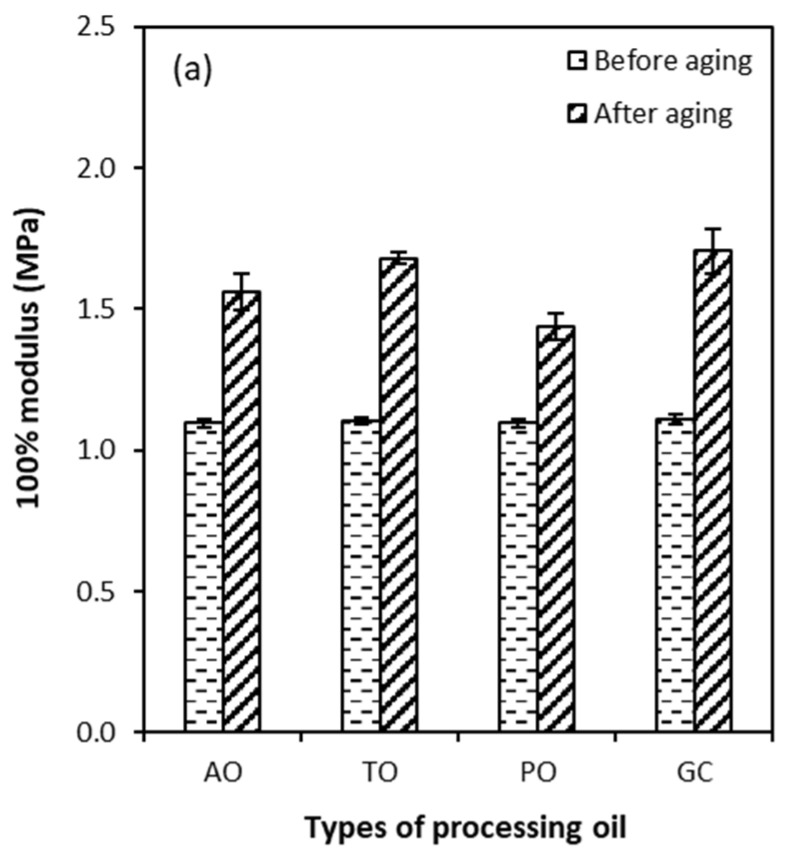
100% modulus (**a**) and 300% modulus (**b**) of the CB-filled NR compounds with different types of processing oil before and after thermal aging.

**Table 1 polymers-15-03599-t001:** Formulations of compounds with varying types of processing oil.

Ingredients	Quantity (phr)			
	AO	TO	PO	GC
RSS#3	100.0	100.0	100.0	100.0
ZnO	3.0	3.0	3.0	3.0
Stearic acid	1.0	1.0	1.0	1.0
CB (N330)	30.0	30.0	30.0	30.0
Aromatic oil	5.0	-	-	-
Paraffinic oil	-	5.0	-	-
TDAE oil	-	-	5.0	-
Glycerol	-	-	-	5.0
CBS	1.0	1.0	1.0	1.0
Sulfur	2.0	2.0	2.0	2.0

**Table 2 polymers-15-03599-t002:** Abbreviations of compound formulations.

Coding	Types of Compounds
AO	CB/aromatic oil-filled NR
TO	CB/treated distillate aromatic extracted oil-filled NR
PO	CB/paraffinic oil-filled NR
GC	CB/glycerol-filled NR

**Table 3 polymers-15-03599-t003:** Mixing sequence of rubber compound via two-roll mill.

Step	Mixing Procedure	Time (min)
1	Mastication of RSS#3	2
2	Add ZnO and stearic acid and continue mixing	7
3	Add CB (N330) and processing oil, continue mixing	18
4	Add CBS and sulfur, continue mixing	6
Total		33

**Table 4 polymers-15-03599-t004:** Solubility parameter value of NR and processing oils [24,25,26,27].

Rubber and Oil Types	Solubility Parameter (MPa^1/2^)
Natural rubber	16.4
Aromatic oil	18.4
Treated distillate aromatic extracted oil	16.7
Paraffinic oil	16.6
Glycerol	34.8

**Table 5 polymers-15-03599-t005:** Mechanical properties of the CB-filled NR compounds with different types of processing oil.

Types of Processing Oil	Tensile Strength (MPa)	Elongation at Break (%)	Hardness (Shore A)	Compression Set (%)
AO	26.0 ± 1.4	694 ± 36.2	55.8 ± 0.4	23.9 ± 1.8
TO	27.8 ± 0.3	748 ± 6.3	56.6 ± 0.8	23.9 ± 0.2
PO	26.2 ± 0.1	720 ± 6.4	55.9 ± 0.2	23.7 ± 1.3
GC	25.6 ± 1.0	666 ± 7.2	57.2 ± 0.5	23.5 ± 1.3

## Data Availability

Not applicable.

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
