# Peer review of "Effect of Glycerol as Processing Oil in Natural Rubber/Carbon Black Composites: Processing, Mechanical, and Thermal Aging Properties"

_polymers, 2023, doi:10.3390/polym15173599_

Round 1

Reviewer 1 Report

The manuscript is required to incorporate the suggestions/comments for a better understanding of the reader.

1. What are the suitable properties highlighted in the abstract?

2. The author did the literature survey up to 2021. It creates doubt about their research gap and novelty. It is suggested to include the literature from 2022 and 2023 and elaborate on your research gap followed by the objective of the work.

3. Author can be included in the set-up and sample images.

4. The author must include the standard deviation in all the figures and also report the repeatability of data in the experiment section.

5. Have the authors performed the crosslinking test on prepared samples? The author has mentioned the statement that higher hardness usually 280 results from an increased amount of crosslinking in rubber matrices. Justify.

6. Where NR is used in this research as 303 the main matrix, the main-chain carbon-carbon double bonds in the chemical structure of 304 NR can easily be oxidized by heat aging, for the reason of which the results after thermal aging 305 are always lower than before. Shorten the sentence for better and clear understanding.

Author Response

Reviewer 1

The manuscript is required to incorporate the suggestions/comments for a better understanding of the reader

1. What are the suitable properties highlighted in the abstract?

ANS: Thank you very much the reviewer for considering the paper. We have revised the abstract into the better scope presenting that can be followed below:

The present work aims to study the effect of glycerol as a replacement for mineral oils in natural rubber (NR) composites to obtain suitable properties through cure characteristics, mechanical properties, and thermal stability. Glycerol was used at a 5 phr in the compound with carbon black as a reinforcing filler and was compared to mineral processing oils such as aromatic oil, treated distillate aromatic extracted oil, and paraffinic oil. Comparing to the other oils, glycerol exhibits better maximum torque and torque differences. Also, shorter scorch time, cure time, and a higher cure rate index of the compounds were observed. However, although the received mechanical properties, including tensile strength, elongation at break, and compression set of the vulcanized rubber using glycerol, show slightly lower values than others, the 100% and 300% moduli, as well as the hardness of the composites filled with glycerol exhibit better values relative to other commercial oil. These findings demonstrate that glycerol overall presents a good balance of properties, making it beneficial to use glycerol as a substitute for mineral oil in tire, shoe sole, and rubber stopper applications.”

2. The author did the literature survey up to 2021. It creates doubt about their research gap and novelty. It is suggested to include the literature from 2022 and 2023 and elaborate on your research gap followed by the objective of the work.

ANS: We do appreciate for the reviewer comments, so that the update references of the relative works are sited as indicated in “BLUE” letters of the manuscript following in the “Introduction” and “References” sections descripting below:

-“Introduction” section

“Moreover, glycerol finds application in various industrial sectors, including the pharmaceutical industry, cosmetics, and personal care items. It serves as a humectant in food, contributes to the production of resins, detergents, plastics, and tobacco, and also functions as a plasticizer”

“Calcium lignosulfonate was introduced into styrene–butadiene rubber (SBR) and acrylonitrile–butadiene rubber (NBR) at a consistent ratio of 30 phr. Glycerol, utilized as a rubber formulation plasticizer, was varied in concentration from 5 to 20 phr. The investigation demonstrated that incorporating glycerol as a plasticizer led to a decrease in the viscosity of the rubber compounds, facilitating improved dispersion and even distribution of the filler within the rubber matrices. This, in turn, enhanced the adhesion and compatibility between the filler and the rubber matrices, resulting in a notable enhancement of tensile properties [13]. An array of compounds derived from glycerol were investigated for their potential as plasticizers in flexible poly(vinyl chloride) (PVC) blends. The influence of the chemical structure of the plasticizers on their performance, migration tendencies, and blend morphology was assessed and juxtaposed with blends employing the commercial plasticizer dioctyl terephthalate (DOTP). Blends containing 40 phr (parts per hundred rubber) of glycerol-based plasticizer exhibited a significant reduction (ranging from 54 to 86°C) in glass transition temperature (Tg) when compared to pure PVC (with a Tg of approximately 80°C). Tensile testing of samples formulated with the glycerol-derived alternatives revealed higher flexibility (elongation at break values reaching up to 97%) in contrast to DOTP (with an elongation at break value of 75%) under equivalent plasticizer contents [15]. Enhancing the hydrophobic surface characteristics of silicone rubber through electron beam irradiation while utilizing a glycerol layer. Despite glycerol's intrinsic hydrophilic nature, the treatment resulted in a surprising 19.9% increase in the contact angle of the silicone rubber. This enhancement in hydrophobicity can be attributed to the formation of a network structure on the silicone rubber surface, despite the notable hydroxyl content [16]. The decrease in elastic modulus as dielectric permittivity rises, combined with escalating glycerol loading was observed in glycerol filled silicone rubber composites, indicates the suitability of this material as a prospective option for soft dielectric sensors and actuator applications [17-18].”

“References” section

9. Yong, M.Y.; Basirun, W.J.; Sarih, N.M.; Shalauddin, Md.; Lee, S.Y.; Ang, D.T.C. Utilization of liquid epoxidized natural rubber as prepolymer and crosslinker in development of UV-curable palm oil-based alkyd coating. React. Funct. Polym. 2023, 105658;

DOI:10.1016/j.reactfunctpolym.2023.105658

13. Kruželák, J.; Hložeková, K.; Kvasniˇcáková, A.; Džuganová, M.; Chodák, I.; Hudec, I. Application of plasticizer glycerol in lignosulfonate-filled rubber compounds based on SBR and NBR. Materials 2023, 16, 635, 1–21;

DOI:10.3390/ma16020635

15. Halloran, M.W.; Nicell, J.A.; Leask, R.L.; Marić, M. Bio-based glycerol plasticizers for flexible poly(vinyl chloride) blends. J. Appl. Polym. Sci. 2022, 139, No. e527;

DOI:10.1002/app.52778

16. Sheng, K.; Dong, X.; Chen, Z.; Zhou, Z.; Gu, Y.; Huang, J. Increasing the surface hydrophobicity of silicone rubber by electron beam irradiation in the presence of a glycerol layer. Appl. Surf. Sci. 2022, 591, 153097;

DOI:10.1016/j.apsusc.2022.153097

17. Mathias, K.A.; Hiremath, S.; Kulkarni, S.M. Experimental studies on mechanical and dielectric behavior of Glycerol filled Silicone rubber composites. Eng. Res. Express. 2021, 3, 035010, 1-7;

DOI:10.1088/2631-8695/ac1450

18. Mazurek, P. S.; Yu, L.; Gerhard, R.; Wirges, W.; Skov, A.L. Glycerol as high-permittivity liquid filler in dielectric silicone elastomers. J. Appl. Polym. Sci. 2016, 133, 44153, 1-27;

DOI:10.1002/app.44153

3. Author can be included in the set-up and sample images.

ANS: Thank you very much the reviewer for the requirement. We added the physical appearance of the glycerol indicating in Figure 1 showing “BLUE” letters in “Material” section.

4. The author must include the standard deviation in all the figures and also report the repeatability of data in the experiment section.

ANS: We have added the repeatability data of each test and also revised the figure with SD indication indicating in “Characterization” and “Results and discussion” sections with “BLUE” letter.

5. Have the authors performed the crosslinking test on prepared samples? The author has mentioned the statement that higher hardness usually 280 results from an increased amount of crosslinking in rubber matrices. Justify.

ANS: We did agree for the comments. However, crosslink density is officially related to the received torque differences of the compound after testing through the vulcanization propagation. Therefore, we revised the sentence in the “Mechanical properties” section as the “BLUE” letters showing below:

“This corresponds to the received of the torque differences showing in Figure 4 since the torque different of the rubber compounds had realized as the estimated crosslink density of the compound after vulcanization formation.”

6. Where NR is used in this research as 303 the main matrix, the main-chain carbon-carbon double bonds in the chemical structure of 304 NR can easily be oxidized by heat aging, for the reason of which the results after thermal aging 305 are always lower than before. Shorten the sentence for better and clear understanding.

ANS: We revised the sentences indicating in “BLUE” letters as can be seen below:

In this case, the NR, which contains carbon-carbon double bonds in the main chain, was used as the main rubber matrix. The thermos-oxidative degradable can be easily occurred, causing lowering thermal properties after providing aging.

Reviewer 2 Report

The manuscript of Naebpetch and co-authors is focused on the effect of glycerol as processing oil in carbon black reinforced natural rubber. Herein, glycerol, aromatic oil, paraffinic oil, and distillate aromatic extracted oil were studied and compared in terms of processing, mechanical, and thermal aging properties. The problem is well posed and well described with an interesting overview of the literature background. The materials' characterization and discussion are comprehensive. I think it is publishable. However, some critical issues should be resolved before consideration for acceptance in this journal. Among others, the compatibility between rubber and glycerol is my main concern. The authors stated that “Non-polar oils are generally used in non-polar rubbers, whereas polar oil is generally used in polar rubber”. However, they are using non-polar NR and glycerol which is polar in nature. This type of system can be unstable, and oil can migrate to the surface of manufactured goods. The authors should comment on this fact.

Moreover, some observations, that the authors may consider, are listed:

1.     Paragraph 3.2., mechanical properties - it is evident that in measurements such as tensile tests multiple samples should be tested to obtain the average value, but in this type of measurement, it is also important to discuss the values of standard deviation. The authors mention an increase in modulus values of 100% and 300% for a glycerol-enriched formula, however, most of these values are within the limit of the standard deviation. This conclusion may be too far-fetched at this point. The reproducibility of the results may also provide additional information on the quality of carbon black dispersion. Please make a relative comment.

2.     The authors conducted Shore A hardness tests but only on samples before thermal aging. Shore A hardness test as well as the results of 100% modulus are omitted in this section. It would be interesting to mention these values, especially since an increased level of crosslinking after thermal aging is postulated.

Minor remarks concerning the presentation of data:

3.     section 3.1 and 3.3 - the presentation of numerical data in tables or even in the form of a note in the figure can significantly improve the readability of the data

Author Response

Reviewer 2

The manuscript of Naebpetch and co-authors is focused on the effect of glycerol as processing oil in carbon black reinforced natural rubber. Herein, glycerol, aromatic oil, paraffinic oil, and distillate aromatic extracted oil were studied and compared in terms of processing, mechanical, and thermal aging properties. The problem is well posed and well described with an interesting overview of the literature background. The materials' characterization and discussion are comprehensive. I think it is publishable. However, some critical issues should be resolved before consideration for acceptance in this journal. Among others, the compatibility between rubber and glycerol is my main concern. The authors stated that “Non-polar oils are generally used in non-polar rubbers, whereas polar oil is generally used in polar rubber”. However, they are using non-polar NR and glycerol which is polar in nature. This type of system can be unstable, and oil can migrate to the surface of manufactured goods. The authors should comment on this fact.

ANS: Thank you the reviewer for the concern. Different polarity and solubility parameter of the NR and glycerol can be the reason of phases separation inside the composites relative to other oils. However, considering in the other hands, the exited polarity inside glycerol molecular structure can provide well interaction to the polar filler surfaces which further prevent the re-agglomeration of the filler particles after mixing process following the thermodynamic theory. Thus, use of the glycerol can help the properties improvement of the NR composites through this modeling of filler dispersion and distribution improvement. In addition, in order to avoid the misunderstanding, we added the new sentences in the “Mooney Viscosity” section indicating in “BLUE” letters showing below:

“In addition, the determination of improved properties in the composites must consider not only the differences in polarity and solubility between NR and glycerol but also the compatibility of glycerol with filler surfaces. Therefore, the utilization of glycerol can also be applied, along with the incorporation of polar fillers, to mitigate the re-agglomeration of fillers after the mixing process, in accordance with the thermodynamic theory effects.”

Moreover, some observations, that the authors may consider, are listed: 

1. Paragraph 3.2., mechanical properties - it is evident that in measurements such as tensile tests multiple samples should be tested to obtain the average value, but in this type of measurement, it is also important to discuss the values of standard deviation. The authors mention an increase in modulus values of 100% and 300% for a glycerol-enriched formula, however, most of these values are within the limit of the standard deviation. This conclusion may be too far-fetched at this point. The reproducibility of the results may also provide additional information on the quality of carbon black dispersion. Please make a relative comment.

ANS: We have added the SD values and also revised the sentences in the “Conclusion” section in “BLUE” letters seeing below:

“- For the mechanical properties: the tensile strength, elongation at break and compression set of the vulcanized rubber using glycerol show just slightly lower values than for the other types of processing oil. Whereas the 100% modulus, 300% modulus and hardness of the glycerol compound shows slightly higher values than for the other types of processing oil.

- As to the thermal aging properties: the tensile strength and elongation at break of all samples after thermal aging were lower than before. While the 100% and 300% modulus of the samples after thermal aging were all better than before. Glycerol took a middle position.”

2. The authors conducted Shore A hardness tests but only on samples before thermal aging. Shore A hardness test as well as the results of 100% modulus are omitted in this section. It would be interesting to mention these values, especially since an increased level of crosslinking after thermal aging is postulated.

ANS: Thank you very much the reviewer for the suggestion. Unfortunately, hardness of the composites after aging has not available to provide. Thus, we designed to add the 100 and 300 %moduli replacing the hardness since both results had showed the same tendency of composites before and after aging propagation, reflecting in additional Figure 10.

Minor remarks concerning the presentation of data:

3. Section 3.1 and 3.3 - the presentation of numerical data in tables or even in the form of a note in the figure can significantly improve the readability of the data

ANS: We have revised all of them to be better readability.

Round 2

Reviewer 1 Report

No Comments